# The Deubiquitinating Enzyme Inhibitor PR-619 Enhances the Cytotoxicity of Cisplatin via the Suppression of Anti-Apoptotic Bcl-2 Protein: In Vitro and In Vivo Study

**DOI:** 10.3390/cells8101268

**Published:** 2019-10-17

**Authors:** Kuan-Lin Kuo, Shing-Hwa Liu, Wei-Chou Lin, Po-Ming Chow, Yu-Wei Chang, Shao-Ping Yang, Chung-Sheng Shi, Chen-Hsun Hsu, Shih-Ming Liao, Hong-Chiang Chang, Kuo-How Huang

**Affiliations:** 1Graduate Institute of Toxicology, College of Medicine, National Taiwan University, Taipei 100, Taiwan; antibody0123@gmail.com (K.-L.K.); shinghwaliu@ntu.edu.tw (S.-H.L.); meow1812@gmail.com (P.-M.C.); 2Department of Urology, College of Medicine, National Taiwan University, and National Taiwan University Hospital, Taipei 100, Taiwan; andy79122@hotmail.com (Y.-W.C.); emilyang719@gmail.com (S.-P.Y.); chanhsun.hsu@gmail.com (C.-H.H.); sanguine444@gmail.com (S.-M.L.); changhong@ntu.edu.tw (H.-C.C.); 3Department of Pathology, National Taiwan University Hospital, Taipei 100, Taiwan; weichou8@gmail.com; 4Graduate Institute of Clinical Medical Sciences, College of Medicine, Chang Gung University, Taoyuan 333, Taiwan; csshi@mail.cgu.edu.tw

**Keywords:** urothelial carcinoma, PR-619, chemotherapy resistance, deubiquitination, Bcl-2, apoptosis

## Abstract

After chemotherapy for the treatment of metastatic bladder urothelial carcinoma (UC), most patients inevitably encounter drug resistance and resultant treatment failure. Deubiquitinating enzymes (DUBs) remove ubiquitin from target proteins and play a critical role in maintaining protein homeostasis. This study investigated the antitumor effect of PR-619, a DUBs inhibitor, in combination with cisplatin, for bladder UC treatment. Our results showed that PR-619 effectively induced dose- and time-dependent cytotoxicity, apoptosis, and ER-stress related apoptosis in human UC (T24 and BFTC-905) cells. Additionally, co-treatment of PR-619 with cisplatin potentiated cisplatin-induced cytotoxicity in UC cells and was accompanied by the concurrent suppression of Bcl-2. We also proved that Bcl-2 overexpression is related to the chemo-resistant status in patients with metastatic UC by immunohistochemistry (IHC) staining. In a xenograft mice model, we confirmed that PR-619 enhanced the antitumor effect of cisplatin on cisplatin-naïve and cisplatin-resistant UCs. Our results demonstrated that PR-619 effectively enhanced the cisplatin-induced antitumor effect via concurrent suppression of the Bcl-2 level. These findings provide promising insight for developing a therapeutic strategy for UC treatment.

## 1. Introduction

Bladder urothelial carcinoma (UC) is the sixth most common cancer in the United States, with approximately 80,470 predicted new cases in 2019 according to the American Cancer Society [1]. Despite radical cystectomy, approximately 50% of cases of high-grade and muscle-invasive bladder UC progress to metastatic diseases. The mainstay of metastatic bladder UC treatment is cisplatin-based chemotherapy [2]. However, chemotherapy provides an approximately 50% response rate, with a 20% five-year survival rate [3,4]. Most patients experience relapse and resultant mortality due to drug resistance [1]. Searching for novel strategies to circumvent chemoresistance to improve the outcomes of metastatic bladder cancer is imperative.

Cisplatin (cis-diamminedichloroplatinum II, CDDP) is the main component of chemotherapy for metastatic UCs. The antitumor mechanism of cisplatin involves crosslinking with purine DNA bases, forming DNA adducts to hinder DNA replication and transcription [2]. DNA damage subsequently activates the network of damage responses that regulate cell cycle arrest, DNA repair, and apoptosis [2]. Several genetic and epigenetic pathways have been reported to contribute to chemotherapy resistance [3,4]. Among them, the suppression of apoptosis plays a critical role in drug resistance. An altered expression pattern of antiapoptotic and proapoptotic proteins has been linked to decreased apoptosis [5,6,7]. Among them, downregulated B-cell lymphoma-2 (Bcl-2) family proapoptotic proteins or upregulated antiapoptotic molecules have been well-explored [4,8,9,10].

The post-translational attachment of ubiquitin is the key determinant of a protein’s fate. The process of ubiquitination on cellular proteins requires three different enzymes: E1 (ubiquitin-activating enzyme), E2 (ubiquitin-conjugating enzyme), and E3 (ubiquitin ligase). Substrates involved in a variety of cellular processes can be conjugated to a ubiquitin molecule to target proteins for proteasomal degradation [5]. Deubiquitinating enzymes (DUBs) are proteases that reverse protein ubiquitination; thus, the process of ubiquitination can be efficiently modulated to maintain the homeostasis of cellular proteins [6]. DUBs represent a large group of enzymes opposing E3 ligase function. Approximately 100 DUBs are encoded in the human genome [7]. DUB regulation has been demonstrated to be a promising target for cancer therapy [8,9,10]. To date, pharmacological studies of DUBs have identified several novel DUB inhibitors or antagonists for potential clinical use [11]. PR-619 has a broad specificity that inhibits multiple deubiquitinating enzymes [12]. PR-619, as a pan-DUB inhibitor, has been reported to be an effective treatment in some cancers [13].

However, the potential antitumor effects of PR-619 on UCs remain unclear. In this study, we investigated the antitumor effects of PR-619 alone and in combination with cisplatin on human UC. Furthermore, we explored the underlying mechanism involving regulation of the Bcl-2 protein.

## 2. Materials and Methods

### 2.1. Cell Culture

The T24 and BFTC-905 cell lines, both derived from a patient with grade III bladder UC, were cultured in RPMI-1640 medium. We used another two cell lines, SV-HUC-1 and RT-4, to investigate the cytotoxic effect of PR-619. SV-HUC-1 cells derived from simian virus 40-transformed were immortalized and non-tumorigenic human urothelial cells. RT4 cells derived from a transitional cell papilloma were low-grade UC cells. RT-4 and SV-HUC-1 cells were cultured in McCoy’s 5a medium and F12 medium, respectively. Culture media were supplemented with 10% heat-inactivated fetal bovine serum, 1 mM sodium pyruvate, and penicillin (100 units/mL)/streptomycin (100 μg/mL) at 37 °C with 5% CO_2_. Culture media and supplements were purchased from Invitrogen (Carlsbad, CA, USA). All cell lines were obtained from the Bioresource Collection and Research Center (Hsinchu, Taiwan).

Cisplatin-resistant UC cells (T24/R) were derived from the parental T24 cells via the continuous exposure to cisplatin after initial dose-response studies of cisplatin (10–40 μM) over 72 h based on a 50% inhibitory concentration (IC_50_), as we described in our previous study [14].

### 2.2. Reagents and Antibodies

PR-619 was obtained from Merck Millipore (Billerica, MA, USA), cisplatin was obtained from clinical preparations of Abiplatin solution (Pharmachemie BV, GA Haarlem, the Netherlands) and ABT-199 was obtained from Abcam (Cambridge, MA, USA). All other chemicals were purchased from Sigma-Aldrich (St. Louis, MO, USA) or Merck Millipore. For Western blot analysis, the antibodies against CHOP (#2895), Bcl-2 (#2827), P21 (#2947), and cleaved-PARP (#5625) were obtained from Cell Signaling Technology (Danvers, MA, USA). The caspase-4 antibody (#M029-3) was purchased from MBL (Woburn, MA, USA); the β-actin (#109639) and glyceraldehyde 3-phosphate dehydrogenase (GAPDH, #100118) antibodies were purchased from GeneTex (Irvine, CA, USA); the phospho-CDK2 (Tyr15) and phospho-Histone H3 (Ser10) antibodies were purchased from Abcam (#136810); the ubiquitin antibody (#BML-PW0150) was purchased form Enzo Life Sciences (Farmingdale, NY, USA), and the other antibodies against GRP78 (#sc-13968) and α-tubulin (#sc-5286) were purchased from Santa Cruz Biotechnology (Santa Cruz, CA, USA). For immunohistochemistry (IHC), the Bcl-2 antibody (#sc-7382) was obtained from Santa Cruz Biotechnology.

### 2.3. Measurement of Cell Viability

Cell viability were analyzed by a 3-(4,5-dimethylthiazol-2-yl)-2,5-diphenyltetrazolium bromide (MTT) assay (Sigma-Aldrich). Briefly, cells were seeded in 96-well microplates (5000 cells/well) and incubated at 37 °C for 24 h. Then, cells were subjected to various treatments for the indicated period and were incubated in medium containing 0.5 mg/mL MTT at 37 °C for 4 h. The reduced crystals were dissolved in DMSO for the measurement of absorbance at 570 nm with the Thermo Scientific Multiskan GO plate reader (Thermo Scientific, Rockford, IL, USA).

### 2.4. Western Blot Analysis

Cells were lysed with lysis buffer (Cell Signaling Technology) of cold phosphate-buffered saline (PBS). The supernatants of the cell lysate were centrifuged at 14,000 rpm for 10 min at 4 °C. The bicinchoninic acid protein assay (Thermo Scientific) was used to detect the total protein concentrations. Equal amounts of proteins obtained from each group and mixed with loading buffer (Biotools, Taipei, Taiwan) were subjected to sodium dodecyl sulfate-polyacrylamide gel electrophoresis, and were then transferred onto polyvinylidene fluoride (PVDF) membranes (Merck Millipore). After blocking with 5% bovine serum albumin (BSA) in PBS, the membrane was incubated with indicated antibodies in PBS at 4 °C overnight. After being washed twice with TBST (tris buffered saline containing 0.05% Tween 20), the membrane was incubated with horseradish-peroxidase-conjugated secondary antibodies (GeneTex) at room temperature for 2 h. The antibody-labeled membrane was washed twice with TBST. Then, enhanced chemiluminescence (ECL) substrates (Merck Millipore and Biotools) were used for detection in an ImageQuant LAS 4000 (GE Healthcare, Chicago, IL, USA) system [15]. Furthermore, the expression levels of target proteins were quantified by Image J software (version 1.52q, NIH, Bethesda, MD, USA) with normalizing to each internal control.

### 2.5. Apoptosis Assay

An apoptosis assay was performed using a Muse^®^ Annexin V and Dead Cell Assay Kit (#MCH100105) in accordance with the manufacturer’s protocol. The stained apoptotic cells were then examined and quantified by a Muse^®^ Cell Analyzer (Luminex Co., Austin, TX, USA) and equipped with the Muse Analysis software (version 1.6.0.0) [16].

### 2.6. Cell Cycle Analysis by Flow Cytometry

Cells were seeded until 40% confluency was reached. Cells were then treated with DMSO (control) or PR-169 for 24 h. The cells were subjected to the Muse^®^ Cell Cycle Assay Kit (#MCH100106) for cell cycle analysis using the Muse^®^ Cell Analyzer and equipped with the Muse Analysis software.

### 2.7. Combination Effect of PR-619 and Cisplatin

The combined effects of PR-619 and cisplatin were determined using the CalcuSyn software (version 1.1.1, 1996, Biosoft, Cambridge, UK). The combination effect was evaluated with the treatment of PR-619 and cisplatin at the ratio of 1:2. The median-effect and combination index (CI) were analyzed as previously described [16,17,18]. CI values of less than one, equal to one, and greater than one were defined as synergistic, additive, and antagonistic, respectively.

### 2.8. Immunohistochemistry (IHC) in Human UC Specimens

Formalin-fixed, paraffin-embedded tissue blocks, and fresh UC specimens from six patients with metastatic bladder UC were collected from patients who had received systemic chemotherapy with gemcitabine and cisplatin. Among them, three were defined as chemo-resistant because of disease progression during chemotherapy, while three were defined as chemo-sensitive due to being responsive to chemotherapy. IHC staining by Bcl-2 antibodies was performed as previously described on 5 μm sections of formalin-fixed, paraffin-embedded specimens. The board-certified uro-oncology specialist pathologist (Lin W.C), who was unaware of the clinical data, evaluated the immunoreactivity of Bcl-2 and determined the IHC score. The staining intensity was categorized as zero (negative), one (weak staining), two (moderate staining), or three (strong staining). The mean percentage of positivity in tumor cells was determined by counting at least 10 random fields at both 40 and 400 magnification. The IHC score was calculated by multiplying the intensity score by the positive staining. Meanwhile, the study involved human participants so was approved by the institutional research ethics committee (No. 201901032RINB).

### 2.9. In Vivo Xenograft Experiments

Cells (5 × 10^5^) were suspended in 100 μL of serum-free media and mixed with an equivalent volume of Matrigel (BD Biosciences). The aforementioned mixture was subcutaneously injected into the dorsal flanks of 6–8 weeks old nude mice (obtained from the Taiwan National Laboratory Animal Center, Taipei, Taiwan). Once the tumors grew to 100–150 mm^3^, mice were intraperitoneally treated with PR-619 (10 mg/kg/day, *n* = 5), cisplatin (10 mg/kg, three times/week, *n* = 5), or the combination of cisplatin with PR-619 (*n* = 5) for three weeks. The tumor sizes were measured using calipers every week. The tumor volume was calculated as follows: Longest tumor diameter × (shortest tumor diameter)^2^/2. Tumors were abscised and photographed. The study involving animal experiments complied with the ARRIVE guidelines and was approved by the National Taiwan University College of Medicine and College of Public Health Institutional Animal Care and Use Committee (IACUC, No. 20180483).

### 2.10. Statistical Analysis

Statistical analyses were performed using the GraphPad Prism 6 software, with all data being presented as means ± standard deviations or standard errors of the means. Data with two groups were analyzed by a two-tailed Student’s *t*-test, and data with multiple groups were analyzed by one-way ANOVA, followed by Bonferroni’s post-hoc test. *p* < 0.05 was considered statistically significant.

## 3. Results

### 3.1. PR-619 Induced Cytotoxicity and Apoptosis in Human UC Cells in a Dose-dependent and Time-Dependent Manner

We first investigated the effects of PR-619 (3–15 μM) on the viability of human UC cells (T24 and BFTC-905) at 24 h, 48 h, and 72 h, respectively. As illustrated in Figure 1A,B, PR-619 effectively induced cytotoxicity and apoptosis in both T24 and BFTC cells in a dose- and time-dependent manner. Additionally, we found that PR-619 induced cytotoxicity in low-grade RT-4 UC cells and cisplatin-resistant UC cells (T24/R) in a dose- and time-dependent manner (Appendix A). We also overserved less cytotoxicity of PR-619 on SV-HUC-1 cell line, which is a neoplastic transformation of SV40-immortalized human urothelial cell line (Appendix A).

### 3.2. PR-619 Induced ER Stress and ER-Stress Related Apoptosis in Human UC Cells

The regulatory mechanisms of apoptosis depend on the balanced action between ubiquitination and deubiquitination systems. DUBs play essential roles in modulating the process of apoptosis. Furthermore, we examined the apoptotic effect of PR-619 (5, 7.5, and 10 μM) on T24 and BFTC-905 cells. Our results show that PR-619 induced polyubiquitination, Bcl-2 downregulation, and concurrent PARP cleavage in a dose-dependent manner (Figure 2A,B).

In addition to the apoptotic effect of PR-619 on UC cells, the endoplasmic reticulum (ER)-stress-related apoptosis proteins (CHOP and caspase-4) increased after PR-619 treatment. Consistently, the ER stress-related chaperon protein, GRP78, increased after PR-619 treatment. We assumed that PR-619 disturbed protein homeostasis of UC cells and induced ER stress, followed by apoptosis in UC cells.

### 3.3. PR-619 Induced G0/G1 Arrest in UC Cells

We examined the effect of PR-619 on the cell cycle progression of human UC cells. Flow cytometry analysis showed that PR-619-treated (7.5 μM) UC cells were blocked in the G0/G1 phase after 24 h (Figure 3A,B). UC cells treated by 7.5 uM PR-619 for 24 h decreased phospho-Histone H3 (Ser10); meanwhile, increased p21, a cyclin-dependent kinase (CDK) inhibitor, and phospho-CDK2 (Tyr15), a G0/G1 arrest marker, in both UC cells. All these results consistently indicated that PR-619 induced G0/G1 arrest in UC cells (Figure 3C).

### 3.4. PR-619 Enhanced Cisplatin-Induced Cytotoxicity in Human UC Cells

We then examined the apoptotic effect of cisplatin (15 μM), PR-619 (7.5 μM), or in combination on T24 and BFTC-905 cells at 24 h. The data are presented in Figure 4. PR-619 in combination with cisplatin significantly potentiated apoptosis compared to that of cisplatin alone (Figure 4A–C). Western blotting showed that PR-619 in combination with cisplatin suppressed the cisplatin-induced overexpression of Bcl-2, an anti-apoptotic regulator. These data consistently indicated that PR-619 potentiated the apoptotic effect of cisplatin on UC cells. We also found that ABT-199, a Bcl-2 inhibitor, elucidated limited impact on the cytotoxicity of PR-619. This data indicated that suppression of Bcl-2 is essential for PR-619-induced cytotoxicity in UC cells (Appendix A). Additionally, T24 and BFTC-905 cells treated with PR-619 in combination with cisplatin at a 1:2 ratio for 24 h showed that PR-619 enhanced cisplatin-induced cytotoxicity (Figure 4G,H).

### 3.5. Overexpression of Bcl-2 Was Associated with Chemotherapy Resistance in Patients with Metastatic UC

We then examined the expression of Bcl-2 in bladder UC tissue samples from six patients with metastatic UC who had received systemic chemotherapy with a gemcitabine and cisplatin regimen by IHC staining. Among them, three chemo-sensitive and three chemo-resistant UC tumors were included. As is shown in Figure 5, the immunoreactivity of Bcl-2 in chemo-resistant UCs (lower part) was stronger compared to that in chemo-sensitive UCs (upper part). The IHC score is shown in Appendix A. Consistent with previous studies, our results supported that Bcl-2 overexpression was associated with chemotherapy resistance in UC [19].

### 3.6. PR-619 Enhanced the Antitumor Effect of Cisplatin on a Cisplatin-Naïve and Cisplatin-Resistant UC Xenograft of Nude Mice

We evaluated the antitumor effects of treatment with cisplatin, PR-619, or combined treatment with cisplatin and PR-619 in vivo by using a xenograft mouse model. T24 and BFTC-905UC cells were mixed with Matrigel and injected subcutaneously into flanks of nude mice. As we described in the Methods section, mice were divided into four groups based on different treatment: DMSO (control, *n* = 5), cisplatin (*n* = 5), PR-619 (10 mg/kg/day, *n* = 5), or cisplatin combined with PR-619 (*n* = 5) for three weeks. Combined treatment with cisplatin and PR-619 showed the most significant antitumor effect on xenograft tumors of both T24 and BFTC-905 compared to single agent (cisplatin or PR-619) treatment (Figure 6A,B). In addition to drug combination treatment for improving the efficacy of chemotherapy, a novel agent for circumventing cisplatin resistance provided other solutions for this clinically unsolved issue. We further examined the antitumor effect of PR-619 on cisplatin-resistant UCs (T24/R) in vitro and in vivo. PR-619 induced cytotoxicity and apoptosis in a dose-dependent manner after 24 h treatment. The in vivo data exhibited using the xenograft mice model showed that PR-619 (10 mg/kg/day) inhibited tumor growth during a 28-day period of treatment (Appendix A).

## 4. Discussion

Cisplatin is still the primary constituent of standard chemotherapeutic regimens for the treatment of metastatic UCs. However, its toxicity and the emergence of drug resistance have compromised its therapeutic efficacy. The use of combined therapies without overlapping toxicities to target alternative pathways to overcome drug resistance or improve therapeutic efficacy has been widely applied in cancer treatment. In this study, we demonstrated that the DUB inhibitor, PR-619, exhibited an antitumor effect on human UCs and cisplatin-resistant UCs. PR-619 enhanced cisplatin-induced cytotoxicity in UCs.

Ubiquitination of proteins plays a key role in signal transduction pathways and mediates protein stability [20]. DUBs remove ubiquitin chains from post-translationally-modified proteins that play a key role in maintaining protein homeostasis. PR-619 was previously shown to exert a broad DUB-inhibitory profile in living cells, resulting in an accumulation of polyubiquitinated proteins and 26S proteasome complexes [21]. Bladder UCs have been confirmed to possess significant genomic instability and a high mutation burden, which made UC cells tend to develop drug resistance followed by selective pressure during anti-cancer treatment.

Nevertheless, bladder UCs are good candidates for DUB inhibitor treatment. Approximately 76% of all primary bladder tumors display mutations in at least one chromatin regulatory gene [22]. In our unpublished data, bladder UC showed high ER stress and an unfolded protein response. Targeting the pathway to disturb protein homeostasis and potentiate ER stress is considered a promising strategy for UC treatment. The present study confirmed our assumption [15,17,23].

Bcl-2 (B-cell lymphoma 2), a member of the Bcl-2 family, regulates cell death, apoptosis, by either inhibiting anti-apoptotic or inducing pro-apoptotic factors [24]. Bcl-2 family proteins also participate in the regulation of many vital cellular functions [25], and play a central role in the regulation of apoptosis machinery and its correct functioning, which is a key element in the effectiveness of current anti-cancer treatments [26]. Bcl-2 over-expression is related to enhanced apoptosis evasion [27] and resistance to chemotherapy [19]. Cancer cell sensitivity to chemotherapy depends on the expression level of anti-apoptotic proteins, as well as the activity of apoptotic pathways in response to death signals [28]. Apoptosis evasion via Bcl-2 or other anti-apoptotic proteins overexpression has become a promising target for new therapeutic strategy [29]. Strategies to inhibit Bcl-2-like proteins has led to the development of a series of small molecules that bind to the hydrophobic cleft of Bcl-2-proteins [30], which are currently under investigation of clinical and preclinical stages.

Our study also supported the phenomenon that PR-619 enhanced cisplatin-induced cytotoxicity with the concurrent suppression of Bcl-2. The aberrant Bcl-2 expression in UC cells after chemotherapy provides a reason to develop Bcl-2-targeting strategies to improve clinical outcomes. In this regard, further study on the evolving field of personalized precision medicine approaches coupled with molecular biology studies focused on Bcl-2-related pathways is warranted.

## 5. Conclusions

In summary, PR-619 enhances the cytotoxicity of cisplatin and effectively inhibits the tumor growth of cisplatin-resistant UC. The aberrant expression of Bcl-2 related to chemotherapy provides a promising target to find out a strategy to improve the therapeutic efficacy and circumvent chemotherapy in human UCs.

## Figures and Tables

**Figure 1 cells-08-01268-f001:**
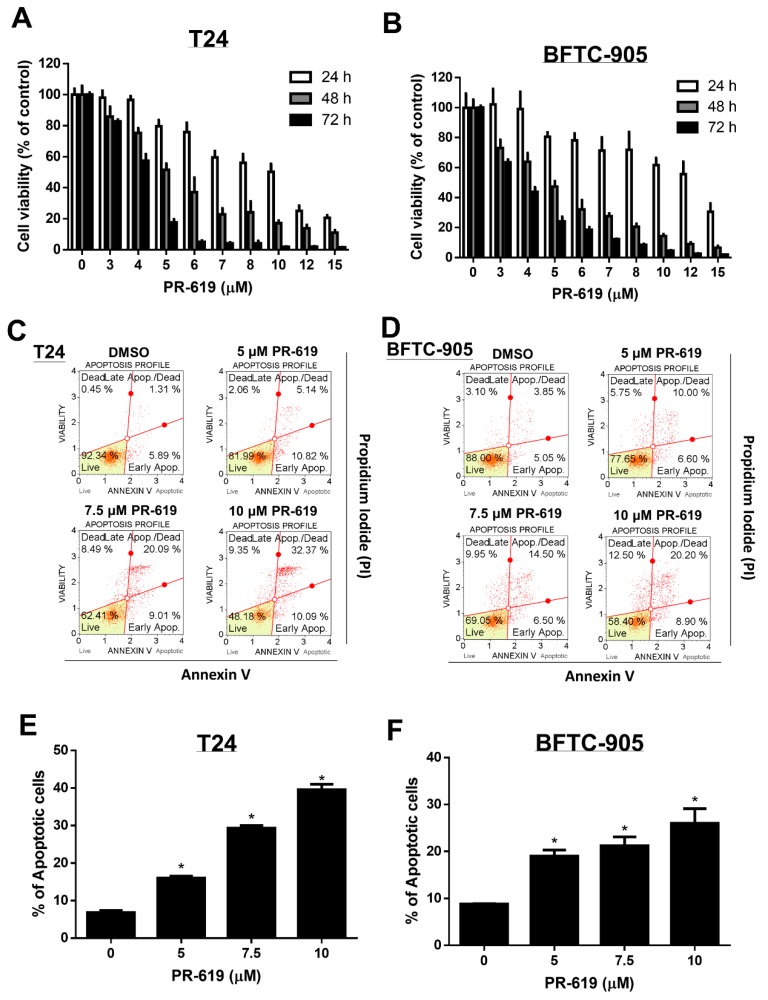
PR-619 induced cytotoxicity and apoptosis in human urothelial carcinoma cells in a dose-dependent and time-dependent manner. (**A**) T24 and (**B**) BFTC-905 cells were treated with various concentrations of PR-619 (3–15 μM) for 24 h, 48 h, and 72 h, respectively. Cell viability was assessed using the 3-(4,5-dimethylthiazol-2-yl)-2,5-diphenyltetrazolium bromide (MTT) assay. (**C**) T24 and (**D**) BFTC-905 cells were exposed to PR-619 (5, 7.5, and 10 μM) or DMSO for 24 h. Apoptotic cells were analyzed through FACS flow cytometry with propidium iodide and annexin V-FITC staining. (**E**,**F**) show the quantitative analyses of apoptosis presented as the means ± SD; * *p* < 0.05 compared with controls. All results shown are representative of at least three independent experiments.

**Figure 2 cells-08-01268-f002:**
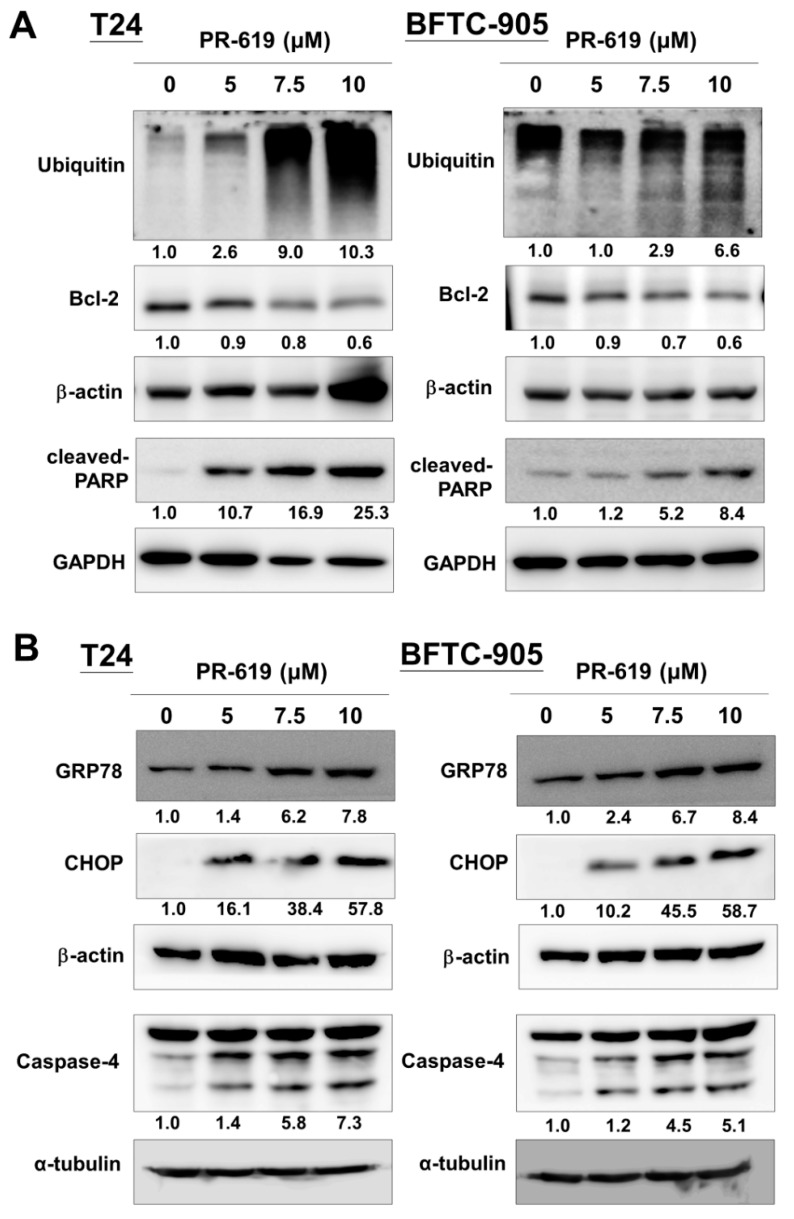
PR-619 induced ER stress and ER-stress-related apoptosis in human urothelial carcinoma (UC) cells. (**A**) T24 and (**B**) BFTC-905 cells were treated with PR-619 (5, 7.5, and 10 μM) for 24 h. Cell lysates were harvested, and the expression of ubiquitin, bcl-2, cleaved-PARP, GRP78, CHOP, and caspase-4 was assessed using Western blot analysis. All results shown are representative of at least three independent experiments.

**Figure 3 cells-08-01268-f003:**
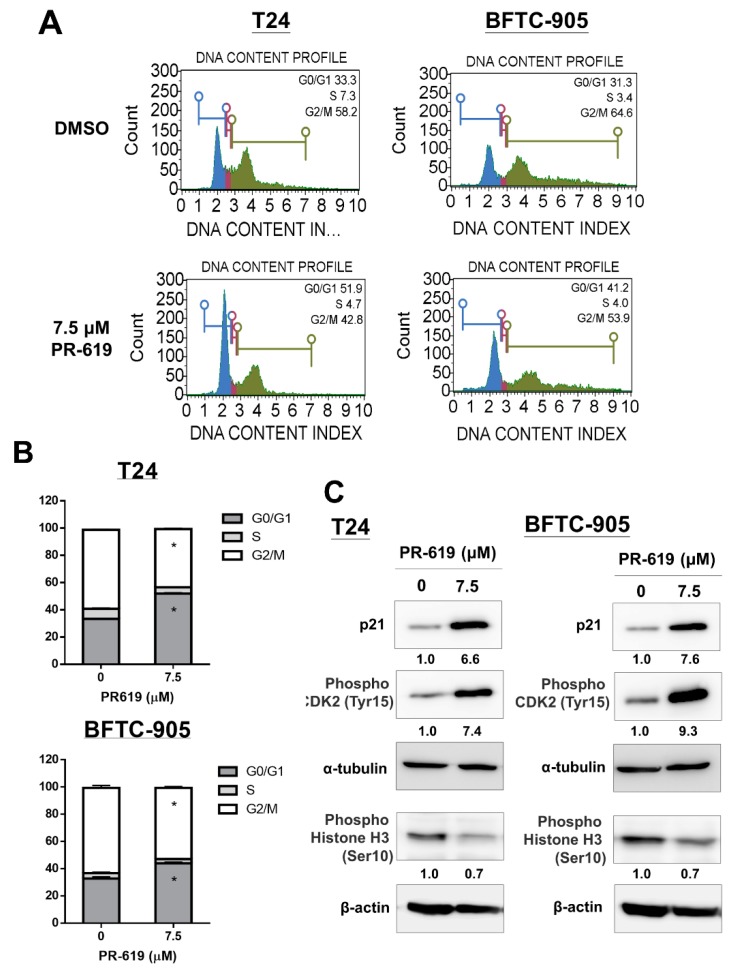
PR-619 induced G0/G1 arrest in human UC cells. (**A**) T24 and BFTC-905 cells were treated with PR-619 (7.5 μM) or DMSO for 24 h. Cell cycle analyses were performed through flow cytometry with PI staining. (**B**) Quantitative data are presented. (**C**) Cells were treated with PR-619 (5, 7.5, and 10 μM) or DMSO for 24 h. Total cell lysates were assessed for p21 using Western blot analysis. Results shown are representative of at least three independent experiments.

**Figure 4 cells-08-01268-f004:**
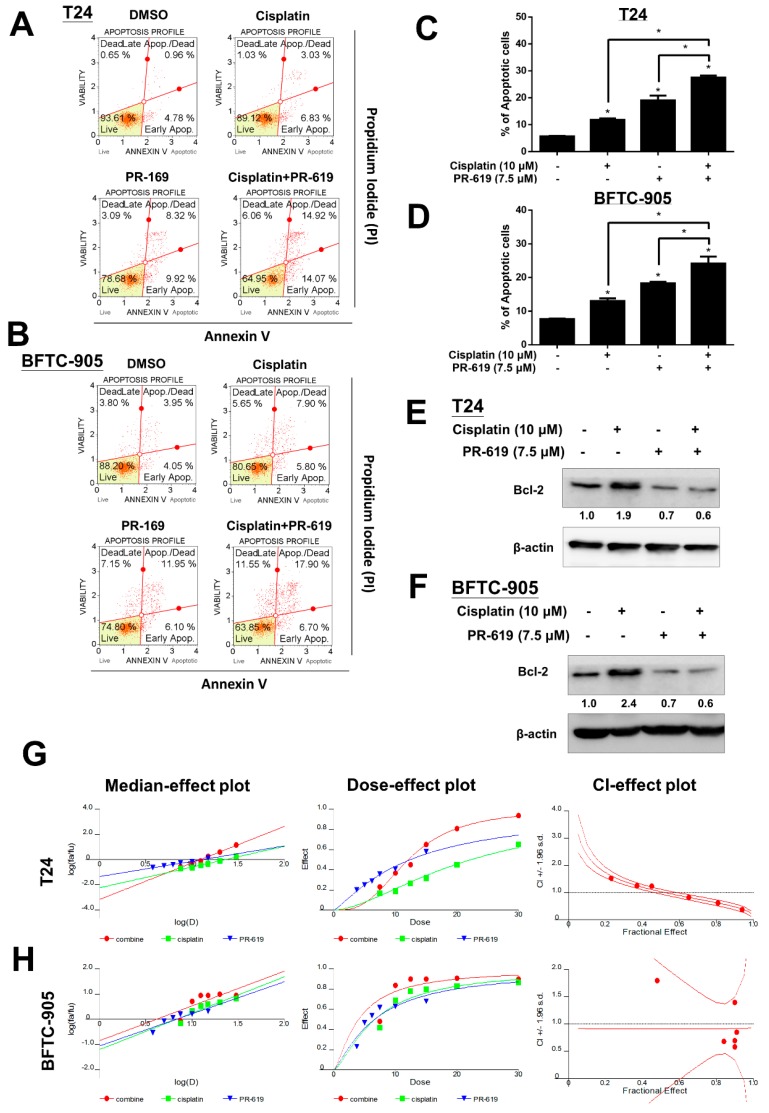
PR-619 enhanced cisplatin-induced cytotoxicity in human UC cells. (**A**) T24 and (**B**) BFTC-905 were treated with PR-619 (7.5 μM), cisplatin (15 μM), or a combination for 24 h. Apoptotic cells were analyzed through FACS flow cytometry with propidium iodide and annexin V-FITC staining. (**C**,**D**) show the quantitative analyses of apoptosis presented as the means ± SD. Data are presented as means ± SD; * *p* < 0.05 compared with controls. All results shown are representative of at least three independent experiments. (**E**) T24 and (**F**) BFTC-905 cells were treated with PR-619 (7.5 μM), cisplatin (15 μM), or a combination for 24 h. Total cell lysates were analyzed for Bcl-2 using Western blot analysis. (**G**) T24 and (**H**) BFTC-905 were exposed to cisplatin in combination with PR-619 at a 2:1 ratio for 24 h. Cell viability was determined using MTT assays. The median-effect plot, dose-effect plot, and combination index (CI) plot of PR-619 in combination with cisplatin were presented. CI values of less than one, equal to one, and greater than one were defined as synergistic, additive, and antagonistic, respectively.

**Figure 5 cells-08-01268-f005:**
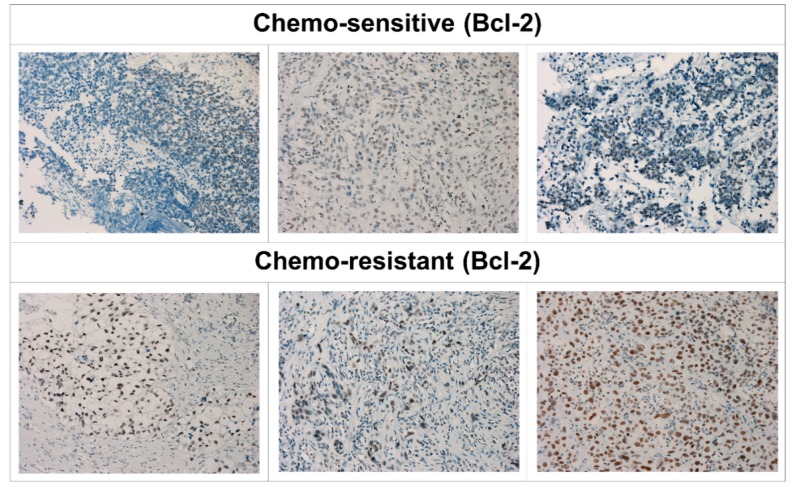
Bcl-2 expression in human metastatic UC cells is associated with chemo-resistance. Imunohistochemical (IHC) staining analysis of Bcl-2 from six patients with metastatic bladder UC, including three (upper part) with a chemo-sensitive and three (lower part) with a chemo-resistant status. IHC staining of formalin fixed, paraffin-embedded bladder UC tissues. The nuclear membrane and cytoplasm of chemo-resistant UC cells show intense staining with antibody to the Bcl-2 protein compared to those in chemo-sensitive UC cells.

**Figure 6 cells-08-01268-f006:**
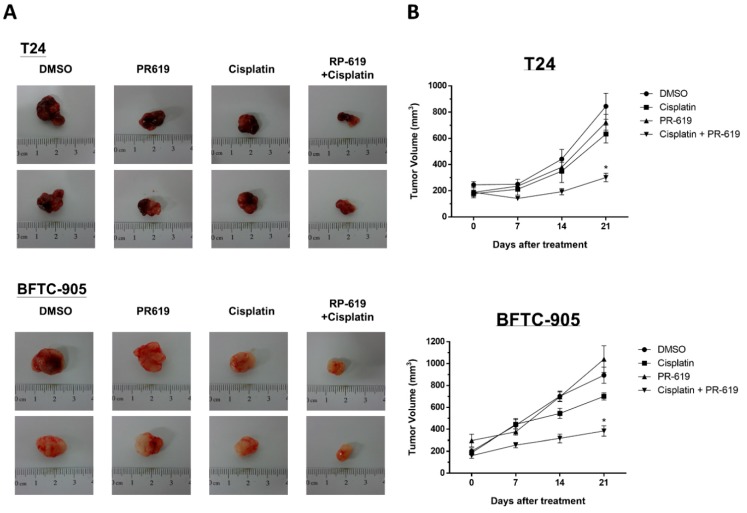
PR-619 enhanced the antitumor effect of cisplatin on a UC xenograft mouse model. Nude mice bearing T24 or BFTC-905 UC tumor xenografts over the bilateral flank area were treated with DMSO (non-treated control, *n* = 5), cisplatin (*n* = 5), PR-619 (*n* = 5), or a combination of cisplatin and PR-619 (*n* = 5) for three weeks. (**A**) Tumor images representing excised tumors from each group. (**B**) Tumor volume for each group during the three-week treatment. The data are presented as means ± standard error of the mean.

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
