# Peer review of "The Deubiquitinating Enzyme Inhibitor PR-619 Enhances the Cytotoxicity of Cisplatin via the Suppression of Anti-Apoptotic Bcl-2 Protein: In Vitro and In Vivo Study"

_cells, 2019, doi:10.3390/cells8101268_

Round 1

Reviewer 1 Report

The authors describe the effects of PR-619 alone and in combination with cisplatin in two UC cell lines. The combination shows more cell death than the treatment with one of the single agents, leading them to state that their effects are synergistic. Furthermore, high Bcl2 protein levels correlate with reduced cell death in urothelial tumors.

This is a mainly observational paper of therapy effects in two tumor cell lines. It is currently unclear whether this is an effect that is specific for cancer cells or that it also applies to normal cells. This is a very important issue, as therapeutic use of a combination of cisplatin and PR-619 will depend on the relative effects on tumor and normal tissue. If toxicity to normal tissue increases similarly or even more than in the tumor, the combination is not expected to give clinical benefit. The paper should provide some evidence for effects on normal cells. Also some additional insight into the mechanisms leading to increased toxicity of the combination would be very useful to predict whether the effect will be tumor-specific (e.g. by analyzing DNA damage induction or repair capacity). At this moment, the manuscript is a random observation in two cell lines.

The second major point is the question whether the effects should be called ‘synergistic’. It is difficult to read figure 4 without proper legends. I assume that the red line in 4G and H is the combination and the two other lines are the single treatments. In that case, both cell lines seem to behave quite differently, with T24 showing a differently shaped curve for the combination than for the single treatments and BFTC-905 with very similar effects of single and combination treatment. Without further mechanistic insight, it is not clear what this means. I would therefore advise not to use the term synergistic in this case. The observation that higher Bcl2 levels reduce cell death is not very novel and does not provide much additional mechanistic insight. It may be worthwhile to mention, but does not solve the mechanism of action.

Minor points:

Don’t use the abbreviation UCs in the abstract without explanation.

Figure 1: baseline apoptosis (20-30%) is very high. Is this an artefact or are there indeed that many apoptotic cells in the culture?

Page 5, line 167 read ‘GRP78 decreased’: the figure shows that it increased. Which is correct?

Figure 3: estimation of S-phase by only studying DNA content is very imprecise. This should be repeated with a protocol involving BrdU incorporation or something similar.

Figure 4: lettering is largely unreadable because of size. Please adapt the figure.

Reviewer 2 Report

While chemotherapy has been established for urothelial carcinoma (UC), in which cisplatin is the primary chemotherapy agent, its treatment is limited due to the risks of developing chemo-resistance and side effects. Combinations of anti tumor drugs are needed to make the regimen more effective. In the manuscript ID cells-583937 entitled, "The deubiquitinating enzyme inhibitor, PR-619, induces synergistic cytotoxicity with cisplatin via suppression of anti-apoptotic Bcl-2 protein: In Vitro and In Vivo Study”, the authors investigated whether a deubiquitination inhibitor PR-619 effectively affects UC. Although this study showed the cytotoxic effects against UC cells, the added insights regarding the underlying mechanisms of action are limited. More detailed analyses are required to be accepted.

Major concerns:
1. The rationale of this drug usage is weak.
The role of DUBs for maintaining protein homeostasis is a fundamental cellular activity. Even normal (non-cancer) cells could be killed by this compound. According to the authors assumption, either proteasome inhibitor (MG132) or ER stress response (UPR) pathway inhibitor (GSK2606414) can be equivalently efficient targets. It is required to provide adequate grounds for targeting DUBs as chemotherapeutic target.

2. The evidence that PR-619 induces cell death through Bcl-2 and ER stress-dependent apoptosis pathways is insufficient.
There is no direct evidence that Bcl-2 or ER stress is involved in the PR-619 induced cell death. To argue that, interventional experiments are essential, which used to be performed in their previous studies (manuscript references 14 and 15). No other factor or pathway was screened, while authors previously reported that BclxL and JNK are involved in (ref15). Based on the results in Fig1 and Fig2, the cell death was induced without ER-stress induction at 5uM.

3. No quality control assures the PR-619 functions as a DUB inhibitor.
The observed cytotoxicity could be a manifestation of unintended off-target effects. At least, the accumulation of poly-ubiquitination should be evaluated. Ideally, the effects of other DUB inhibitors could also be tested.

4. Synergic effect or additive effect?
The effect of PR-619 seems to be limited in the cisplatin-resistant T24R (FigS2). Although the combination index was analyzed, this may suggest the mechanism of PR-619 action is shared with cisplatin. How the authors interpret the contrary findings need to be discussed.

5. Poor figure resolutions and unfriendly results explanations.
There are ample numbers of figure panels, legends and the descriptions in the manuscript looked scientifically insufficient:

All flow cytometry charts and CI plots have low resolution. Please replace to the ones with higher resolution. (Fig1, Fig2, Fig3, Fig4)

Fig1A&B and other viability assay data should be presented as bar graph rather than line chart. If you need to use line chart, x axis would be replaced by time or the scale should evenly follow the concentration (0, 1, 2, 3…,13, 14, 15uM).

The data look not match between Fig1CD and EF.

Fig4C&D, since these are 2x2 factorial experiments, one-way ANOVA would be appropriate rather than t-test.

The interpretations and descriptions for Figure 5 sounds not fully scientific. Some way to quantify the Bcl-2 expression, such as intensity analysis by ImageJ, should be considered. Inserting scale bar will be appreciated. More details should be provided in the methods section.

According to the manuscript (Page 8 Line 231), FigS2 seems in vivo (xenograft) study. But, it looks in vitro results. Overall figure legends and results descriptions are too brief and confusing.

Round 2

Reviewer 1 Report

The major points have been answered partially. 

The first point was the absence of a normal urothelial cell control. The authors now included an experiment with normal urothelial cells. However, they only did a survival for the 24-hour time point and not 48 or 72 hours as for the other cell lines. Furthermore, they failed to do the higher concentrations up to 15 micromolar. The difference in response in the current setting is very small between normal and tumor cells, so a complete analysis is important to investigate this aspect in sufficient detail to draw conclusions about tumor cell specificity.

The second point is the better mechanistic understanding. The authors argue that a major new element is the increased Bcl2 expression in cisplatin resistant human tumors. Figure S4 shows a quantification, but without statistics. It is currently unclear whether there is a significant difference between sensitive and resistant tumors and therefore, this conclusion cannot be drawn from the presented data. The issue of decreased Bcl2 expression upon PR-619 treatment of the cell lines should also be backed up by quantification of the Western blotting. The example shown for T24 shows beta-actin to go up, while Bcl-2 and GAPDH go down. Variability should be quantified.

Finally, the cell cycle analysis did not improve: the same technique (only PI staining) was used to estimate the various cell cycle phases. This needs to be improved before any definitive conclusions can be drawn.

Reviewer 2 Report

The manuscript looks substantially improved. The selective cytotoxicity is no doubt. However, the data are still incomplete to address the question which pathway is responsible for the DUB inhibitor-induced cytotoxicity. The authors should address the following concerns otherwise refocus on the efficacy of DUB inhibitors.

If the suppression of Bcl2 is essential event for this, forced downregulation of Bcl2, by using siRNA or ABT-199 (venetoclax) BCL2 inhibitor, in combination with PR619 treatment should no or limited impact on the cytotoxicity.

The cytotoxicity is already significant at 5uM dose in Figure1. At the same dose, the accumulations of poly-ubiquitin and ER stress markers are unclear in Figure2. Showing longer exposure images of ER stress markers and the quantifications of western blotting analyses will help to discuss this concern. If the protein expression levels show no difference compared with the ones of control, ER stress may not be responsible for the toxicity.

It looks substantially improved as far as I browsed through the revision. My concerns can be ignored by changing the title and revising the abstract, if the manuscript and the perspective of this journal focus on the therapeutic effects rather than the underlying mechanisms.

Round 3

Reviewer 1 Report

The response to my points 1 and 2 are sufficiently addressed now. For
point 3 I still think that the cell cycle analysis is suboptimal. Their
conclusion that fewer cells are cycling upon incubation with their
compound is most probably correct. However, the designation of S and G2
phase cells does not look credible (figure 3A/B). Their S phase
contribution should be much higher, which is caused by placing the
boundaries arbitrarily in the wrong place. I would still suggest to do
one experiment in which they do a BrdU incorporation or EdU
incorporation assay to determine the percentage of S versus G2/M cells.
It will not affect their main conclusion, but it will make the analysis
much better. Furthermore, I fully agree with accepting this paper

Reviewer 2 Report

The authors addressed my concern. With the ABT-199 results, I am convinced that Bcl2 plays some role in this context. I would recommend to accept this manuscript.